# Generalizability of Adversarial Robustness Under Distribution Shifts

**Kumail Alhamoud**[*]                                                 *kumail.hamoud@kaust.edu.sa*
*King Abdullah University of Science and Technology (KAUST)*

**Hasan Abed Al Kader Hammoud**[*]                    *hasanabedalkader.hammoud@kaust.edu.sa*
*King Abdullah University of Science and Technology (KAUST)*

**Motasem Alfarra**                                                *motasem.alfarra@kaust.edu.sa*
*King Abdullah University of Science and Technology (KAUST)*

**Bernard Ghanem**                                                 *bernard.ghanem@kaust.edu.sa*
*King Abdullah University of Science and Technology (KAUST)*

**Reviewed on OpenReview:** *https://openreview.net/forum?id=XNFo3dQiCJ*

## Abstract

Recent progress in empirical and certified robustness promises to deliver reliable and deployable Deep Neural Networks (DNNs). Despite that success, most existing evaluations of DNN robustness have been done on images sampled from the same distribution on which the model was trained. However, in the real world, DNNs may be deployed in dynamic environments that exhibit significant distribution shifts. In this work, we take a first step towards thoroughly investigating the interplay between empirical and certified adversarial robustness on one hand and domain generalization on another. To do so, we train robust models on multiple domains and evaluate their accuracy and robustness on an unseen domain. We observe that: (1) both empirical and certified robustness generalize to unseen domains, and (2) the level of generalizability does not correlate well with input visual similarity, measured by the FID between source and target domains. We also extend our study to cover a real-world medical application, in which adversarial augmentation significantly boosts the generalization of robustness with minimal effect on clean data accuracy.

## 1 Introduction

Deep Neural Networks (DNNs) are vulnerable to small and carefully designed perturbations, known as adversarial attacks (Szegedy et al., 2014; Goodfellow et al., 2015). That is, a DNN $f_\theta : \mathbb{R}^d \to \mathcal{P}(\mathcal{Y})$ can produce two different predictions for the inputs $x \in \mathbb{R}^d$ and $x + \delta$, although both $x$ and $x + \delta$ are perceptually indistinguishable. Furthermore, DNNs are found to be brittle against simple semantic transformations such as rotation, translation, and scaling (Engstrom et al., 2019).

These observations raise concerns regarding the deployability of DNNs in security-critical applications, such as self-driving and medical diagnosis (Papernot et al., 2016; Finlayson et al., 2019; Ma et al., 2021). This brittleness motivated several efforts to build models that are not only accurate but also *robust* (Gu & Rigazio, 2015). Building robust models is usually achieved either **(i)** *empirically*, where the DNN training routine is changed to include such malicious adversarial examples in the training set (Madry et al., 2018), or **(ii)** *certifiably*, where theoretical guarantees are given about the robustness of a DNN in a region around a given input $x$ (Lécuyer et al., 2019). While recent work on adversarial robustness has made significant strides in developing accurate and robust models, most methods are only evaluated on *in-distribution* data. This means

---

[*]The first two authors contributed equally: each author has the right to list their name first on their CV.

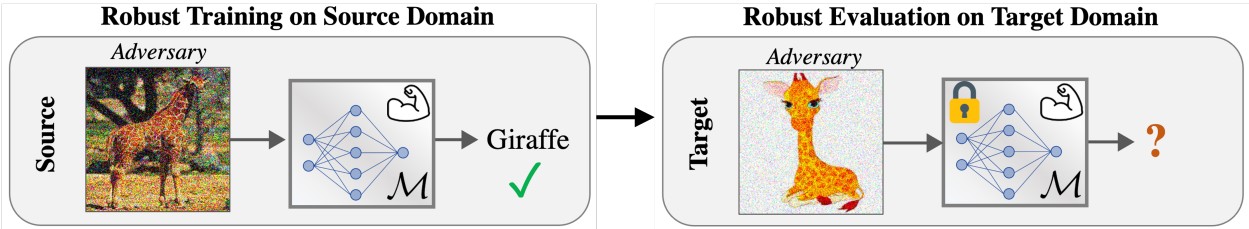

Figure 1: **Does a robust model trained in a (source) domain maintain its robustness when deployed in another (target) domain?** We investigate the generalizability of empirical and certified robustness to various unseen domains.

that the training and testing datasets are assumed to be independently and identically distributed (IID). However, this IID assumption rarely holds in practice, as data in the real world can be sampled from various distributions with significant domain shifts. For example, a deep-learning based medical image classifier may be trained on data collected from one hospital, but later deployed in a different hospital (Bándi et al., 2019). Unfortunately, DNNs struggle to generalize to out-of-domain data (Geirhos et al., 2020; 2021), even in the absence of adversarial examples. This lack of generalization has led the research community to invest in the problem of Domain Generalization (DG). The aim of DG is to learn invariant representations from diverse distributions of data, denoted as *source* domains, such that these representations generalize to an unseen distribution, known as the *target* domain (Wang et al., 2021; Gulrajani & Lopez-Paz, 2021). This setup mimics the unexpected nature of real-world distribution shifts, where models can be regularly exposed to novel domains, and fine-tuning on all these domains becomes impractical. While there has been considerable effort in improving the generalizability of DNNs (Tzeng et al., 2014; Sun & Saenko, 2016; Motiian et al., 2017; Zhang et al., 2021; Shen et al., 2021; Wang et al., 2021; Zhou et al., 2022), the generalizability of adversarial robustness to unseen domains remains unexplored.

Our work examines the interplay between domain generalization and adversarial robustness through comprehensive experiments on five standard DG benchmarks provided by DomainBed (Gulrajani & Lopez-Paz, 2021) and WILDS (Koh et al., 2021). We investigate empirical and certified robustness against input perturbations and spatial deformations. We first investigate the generalizability of empirical robustness, which a DNN obtains by employing the popular adversarial training method (Madry et al., 2018) on the source data. We then inspect the generalizability of certified robustness against input perturbations and parametric deformations by employing Randomized Smoothing (RS) (Cohen et al., 2019) and DeformRS (Alfarra et al., 2022a). To the best of our knowledge, we provide the first large-scale experimental analysis of the generalizability of adversarial robustness to unseen domains. Our analysis leads to the following contributions:

1. We contrast the behavior of robustness under both transfer learning and domain generalization. Unlike transfer learning, domain generalization does not necessarily improve through robust training.

2. We empirically show that visual similarity between the source and target domains does not correlate well with the level of generalizability to the target domain.

3. We show that empirical and certified robustness generalize to unseen domains in different setups, including in a medical setting with a real-world distribution shift.

## 2   Related Work

**Domain Generalization.**   Domain generalization (DG) studies the ability to learn representations that can be readily applied to data from unseen domains (Wang et al., 2021; Zhou et al., 2022). In the DG setup, a model is trained on multiple source domains and then evaluated on an unseen target domain, which exhibits a significant shift from the training domains. DG methods can be categorized into different groups. (i) Data augmentation techniques learn generalizable models by increasing the diversity of the source data (Gong et al., 2019; Zhou et al., 2020; 2021). (ii) Representation learning methods extract domain-invariant

Table 1: **Comparison between Domain Generalization (DG) and Transfer Learning.** DG differs from transfer learning in two ways. 1) The model in DG never sees the target data during training, so fine-tuning on the target is not allowed. 2) Target labels are fixed in DG; however, the target samples are drawn from a domain that is distinct from the source domains.

| Problem Setup | Training Data | Target Data | Problem Condition | Access to Target |
|---|---|---|---|---|
| Transfer learning | $S_{source}, S_{target}$ | $S_{target}$ | $\mathcal{Y}_{source} \neq \mathcal{Y}_{target}$ | ✓ |
| Domain generalization | $S = \{S_i \mid i = 1, \ldots, N\}$ | $S_{N+1}$ | $\mathbb{P}_{XY}(S_k) \neq \mathbb{P}_{XY}(S_n)$ for $k \neq n$ | ✗ |

representations that seamlessly apply in any unseen domain (Blanchard et al., 2011; Nguyen et al., 2021; Lu et al., 2022) (iii) Learning-strategy approaches may achieve generalization through meta-learning, self-supervised learning, or optimization procedures that seek flat minima (Li et al., 2018; Carlucci et al., 2019; Cha et al., 2021). In this work, we study DG from an adversarial robustness lens. In particular, we analyze both the generalization accuracy and robustness of adversarially trained classifiers in unseen domains.

**Adversarial Robustness.** Adversarial attacks are imperceptible, semantic-preserving perturbations that can fool DNNs (Goodfellow et al., 2015; Szegedy et al., 2014). Given the security concerns that adversarial attacks induced, several works proposed changing the training routine to enhance model robustness (Zhang et al., 2019). For example, adversarial training (Madry et al., 2018) encourages the model to classify adversarial examples correctly. While empirical defenses like adversarial training are effective in enhancing the robustness of the underlying model, such approaches do not guarantee robustness. Subsequently, many empirical defenses were broken when more powerful attacks were designed (Carlini & Wagner, 2017; Athalye et al., 2018). As a result, there has been a growing interest in certifiably robust classifiers, for which no adversary can exist in a specified region around a data point (Raghunathan et al., 2018; Mohapatra et al., 2020; Lee et al., 2021). A scalable approach to achieving certified robustness is Randomized Smoothing (RS) (Cohen et al., 2019). RS constructs a smooth classifier from any arbitrary base classifier by outputting the most probable class when the input is subjected to Gaussian noise. Recently, DeformRS extended RS to provide certified robustness against parameterized geometric deformations (Alfarra et al., 2022a; S. et al., 2022). In this work, we aim to study the interplay between (empirical and certified) robustness and domain generalization by deploying adversarial training, RS, and DeformRS.

**Adversarial Training in Dynamic Environments.** To learn generalizable knowledge, machine learning researchers have proposed several problems, such as transfer learning, continual learning, domain adaptation, and domain generalization (Zhuang et al., 2020; Delange et al., 2021; Wang & Deng, 2018; Wang et al., 2021). Among these problems, only transfer learning, where a model pre-trains on tasks with large datasets and then adapts to downstream tasks with limited data, has been thoroughly studied under the lens of adversarial robustness. Salman et al. (2020) showed that, in terms of downstream task accuracy, adversarially trained representations outperform nominally trained representations. Utrera et al. (2021) further explained that adversarial training in the source domain increases shape bias, resulting in better transferability. Finally, Deng et al. (2021) provided a theoretical justification to support these empirical findings. Besides downstream task accuracy, Shafahi et al. (2020) studied the transferability of robustness itself. Although useful, these transfer learning results presume fine-tuning on the target domain, which is not possible in many real-life scenarios. Table 1 illustrates the differences between transfer learning and domain generalization, which is the setup we adopt. In this paper, we take a first step to empirically investigate whether adversarial training leads to robust representations that generalize well without requiring prior knowledge of the target domain.

## 3 Background on Domain Generalization

**Domain Generalization Setup.** Given an input space $\mathcal{X}$ and a label space $\mathcal{Y}$, one can define a joint distribution $\mathbb{P}_{XY}$ over $\mathcal{X}$ and $\mathcal{Y}$. A domain, or distribution, is a collection of samples drawn from $\mathbb{P}_{XY}$. In multi-source domain generalization, there are $N$ source domains of varying sizes $\{D_n\}_{n=1}^N$ where for each $n$, the domain is defined by $D_n = \{(x_j, y_j)\}_{j=1}^{|D_n|} \sim \mathbb{P}_{XY}^{(n)}$. We define the training set $S$ by the union of

the $N$ source domains $S = \bigcup_{n=1}^{N} D_n$, and we assume the existence of some unseen target domain $D_{N+1} = \{(x_j, y_j)\}_{j=1}^{|D_{N+1}|} \sim \mathbb{P}_{XY}^{(N+1)}$. We enforce that $\mathbb{P}_{XY}^{(k)} \neq \mathbb{P}_{XY}^{(n)}$ for $k \neq n, k, n \in \{1, \ldots, N+1\}$, which means that the target domain is distinct from the source domains, which are, in turn, distinct from each other. The aim of DG is to use the source domains $S$ to learn a mapping $f : \mathcal{X} \to \mathcal{Y}$ that minimizes the error on the unseen target domain. More formally, we seek a parameterized model $f_{\theta*}$ such that:

$$\theta^* = \arg\min_{\theta} \mathbb{E}_{(x,y) \sim \mathbb{P}_{XY}^{(N+1)}} \left[ \mathcal{L}(f_\theta(x), y) \right], \tag{1}$$

where $\mathcal{L}$ is the cross-entropy loss for the classification task. However, the model is not allowed to sample the target domain during training. Therefore, most methods use the empirical risk of the source datasets as a proxy for the true target risk. The supervised average risk ($\mathcal{E}$) is given by:

$$\mathcal{E} = \frac{1}{N} \sum_{n=1}^{N} \frac{1}{|S_n|} \sum_{i=1}^{|S_n|} [\mathcal{L}(f_\theta(x), y)] \tag{2}$$

with $(x, y) \sim S$. In practice, we define a fixed held-out validation set $S^v \subset S$. The average risk on this source validation set is used to select the best model, which is evaluated on the target domain *without* any fine-tuning steps. In what follows, Section 4 (and Section 5) investigates the generalizability of empirical (and certified) robustness to diverse target domains. In both sections, we follow the advice of DomainBed (Gulrajani & Lopez-Paz, 2021) in using empirical risk minimization (ERM) as a baseline method, and we focus on the effect of robustifying ERM.

## 4 Empirical Robustness and Domain Generalization

In this section, we study the generalizability of empirical robustness methods that enhance the adversarial robustness of DNNs. We begin with a brief introduction of Adversarial Training (AT) (Madry et al., 2018), after which we study the effect of deploying AT in a domain generalization setup.

### 4.1 Background and Setup

**Adversarial Attacks.** Adversarial attacks are imperceptible perturbations that, once added to a "clean" input sample, cause the classifier $f_\theta$ to misclassify the perturbed sample. Formally, let $(x, y)$ be an input label pair where $f_\theta$ correctly classifies $x$ (*i.e.* $\arg\max_i f_\theta^i(x) = y$). An attacker crafts a small perturbation $\delta$ such that $\arg\max_i f_\theta^i(x + \delta) \neq y$, which is usually obtained by solving the following optimization problem:

$$\max_{\delta} \ \mathcal{L}(f_\theta(x + \delta), y) \qquad \text{s.t. } \|\delta\|_p \leq \epsilon, \tag{3}$$

where $p \in \{2, \infty\}$, $\epsilon > 0$ is a small constant that enforces the imperceptibility of the added perturbation, and $\mathcal{L}$ is a suitable loss function (*e.g.* Cross Entropy). Let $\delta^*$ be the solution to the problem in Eq. 3, then the adversarial example is denoted by $x_{adv} = x + \delta^*$.

**Adversarial Training as Augmentation.** Adversarial Training (AT) (Madry et al., 2018) trains the classifier on adversarial examples rather than clean samples. In particular, AT obtains the network parameters $\theta^*$ by solving the following optimization problem:

$$\min_{\theta} \ \mathbb{E}_{(x,y) \sim \mathcal{D}} \left[ \max_{\delta, \|\delta\|_p \leq \epsilon} \mathcal{L}(f_\theta(x + \delta), y) \right], \tag{4}$$

where $\mathcal{D}$ is a data distribution. In general, the inner maximization problem is solved through $K$ steps of Projected Gradient Descent (PGD) (Madry et al., 2018). While conducting adversarial training enhances the model's robustness against adversarial attacks, this usually comes at the cost of losing some clean accuracy (performance on unperturbed samples). To alleviate the drop in performance, we follow the method by

Zhang et al. (2019) and deploy adversarial training as a data augmentation scheme. In particular, we obtain network parameters $\theta^*$ that minimize the following objective:

$$\theta^* = \arg\min_{\theta} \ \mathbb{E}_{(x,y)\sim\mathbb{P}_{XY}^{(N+1)}}[\lambda\mathcal{L}(f_\theta(x), y) + (1 - \lambda)\mathcal{L}(f_\theta(x_{adv}), y)], \tag{5}$$

where $\lambda \in [0, 1]$ controls the robustness-accuracy trade-off. Furthermore, we experiment with the more powerful method TRADES (Zhang et al., 2019). In particular, TRADES minimizes the following loss:

$$\theta^* = \arg\min_{\theta} \ \mathbb{E}_{(x,y)\sim\mathbb{P}_{XY}^{(N+1)}}[\mathcal{L}(f_\theta(x), y) + \max_{\delta,\|\delta\|_p\leq\epsilon} \beta\mathcal{L}(f_\theta(x), f_\theta(x + \delta))]. \tag{6}$$

**Adversarial Training Setup.** In our experiments, we focus on image classification and adopt the framework of DomainBed (Gulrajani & Lopez-Paz, 2021), which is the standard benchmark in the image domain generalization literature. All models are initialized with a ResNet-50 backbone pre-trained on ImageNet-1K (Deng et al., 2009). We train PGD models with adversarial augmentation to minimize the objective in Eq. 5 on the source domains, where $\lambda = 0.5$ and $x_{adv}$ is computed with a Projected Gradient Descent (PGD) attack (Madry et al., 2018) using 5 PGD steps. The TRADES models are trained to minimize the objective in Eq. 6, where $\beta = 3$. For all the models, the target domain remains unseen until test time. Specifically, we follow the training-domain validation strategy described in DomainBed for model selection. We study robustness under a variety of datasets: PACS, OfficeHome, VLCS, and TerraIncognita (Gulrajani & Lopez-Paz, 2021). In the main paper, we report $\ell_\infty$ results using $\epsilon = 2/255$. The appendix includes additional results for $\epsilon = 4/255$ and $\epsilon = 8/255$ robustness evaluation, leading to similar trends and conclusions.

**Evaluation Setup.** For each considered dataset, we select a subset of $N - 1$ domains to be the source (training) domains and keep the $N^{th}$ domain as the target (evaluation) domain. For example, we have four splits in PACS: Photo, Art, Cartoon, and Sketch. Hence, we have four combinations of source vs. target domain splits. We follow DomainBed in reporting the average result across all N different source vs. target splits Furthermore, we run each experiment with 3 different seeds and report the standard deviation across our runs. Note that we split the source domains (training set) into two subsets: a training subset (80%) and a validation subset (20%). It's important to note that the results we present in Table 2 are based on the validation subset.

To evaluate the robustness of our models, we assessed their performance on the same norm budget they were trained on. For AutoAttack, we used the default number of steps provided in its implementation. For PGD, we conducted the evaluation with 20 steps.

### 4.2 Generalization of Empirical Robustness

In this section, we investigate the generalizability of empirical robustness to unseen domains. More precisely, we are interested in understanding the interplay between standard accuracy and robust accuracy in the scope of source vs. target domains. In Table 2, we report the standard accuracy, $\ell_\infty$ AutoAttack robust accuracy (Croce & Hein, 2020), and $\ell_\infty$ PGD robust accuracy for nominally- and adversarially-trained models using both TRADES Eq. equation 6 and PGD augmentation Eq. equation 5 techniques. The reported results show the average and standard deviation of four runs with different seeds and with $\epsilon$ fixed to $2/255$. Results for $\epsilon = 4/255$ are in the appendix. Next, we analyze these results to answer the following questions.

**Q1: Does adversarial training result in a better generalization to clean samples in the target domain?** No, which is evident from comparing the clean target accuracy of the baseline with the other two adversarial training methods in Table 2. We further observe that the more robust TRADES generalizes worse than the simpler training with PGD augmentation. ❶ **Unlike transfer learning, where robust training in the source domain is favorable, robust training does not improve the clean data accuracy in the target domain if no fine-tuning is allowed.** This result contrasts with findings from the transfer learning literature, where models trained robustly in the source domain outperform standard-trained models across a variety of downstream tasks (Salman et al., 2020). It is especially surprising given that previous works suggest that robust training encourages shape bias over texture bias, hinting at better generalization (Geirhos et al., 2019; Utrera et al., 2021). Moreover, Deng et al. (2021) showed that adversarial

Table 2: **Evaluation of $\ell_\infty$ Robustness.** We train models in the source domain and then evaluate their clean accuracy and robust accuracy in the source and target domains. The models are trained in one of three ways: nominal training (baseline), PGD adversarial training, or TRADES adversarial training. The robust accuracy is measured against AutoAttack and PGD adversarial attacks. For adversarial training and evaluation, the perturbation norm is set to $\epsilon = 2/255$. Overall, the robustness of PGD-trained and TRADES-trained models transfers to the target distribution.

| Method | Dataset | Source | | | Target | | |
| | | Clean Acc. | Acc. (AA) | Acc.(PGD) | Clean Acc. | Acc. (AA) | Acc. (PGD) |
|---|---|---|---|---|---|---|---|
| Baseline | PACS | $94.69 \pm 0.23$ | $1.56 \pm 0.33$ | $3.96 \pm 0.56$ | $80.24 \pm 1.72$ | $0.34 \pm 0.16$ | $1.16 \pm 0.38$ |
| | OfficeHome | $77.08 \pm 0.30$ | $0.03 \pm 0.02$ | $0.40 \pm 0.05$ | $58.88 \pm 0.85$ | $0.00 \pm 0.00$ | $0.56 \pm 0.21$ |
| | VLCS | $84.34 \pm 0.12$ | $0.00 \pm 0.00$ | $0.00 \pm 0.00$ | $74.55 \pm 1.04$ | $0.00 \pm 0.00$ | $0.02 \pm 0.04$ |
| | TerraIncognita | $86.72 \pm 0.22$ | $0.00 \pm 0.00$ | $0.00 \pm 0.00$ | $44.10 \pm 2.52$ | $0.00 \pm 0.00$ | $0.00 \pm 0.00$ |
| PGD | PACS | $92.83 \pm 0.22$ | $76.00 \pm 0.33$ | $78.12 \pm 0.37$ | $75.29 \pm 0.56$ | $56.69 \pm 0.91$ | $55.82 \pm 1.74$ |
| | OfficeHome | $72.04 \pm 0.40$ | $52.32 \pm 0.69$ | $52.81 \pm 0.98$ | $52.19 \pm 0.73$ | $34.03 \pm 0.26$ | $34.51 \pm 0.21$ |
| | VLCS | $79.70 \pm 0.22$ | $58.76 \pm 0.41$ | $60.02 \pm 0.75$ | $69.15 \pm 0.74$ | $47.12 \pm 0.69$ | $46.73 \pm 1.64$ |
| | TerraIncognita | $71.62 \pm 0.74$ | $52.85 \pm 2.25$ | $56.05 \pm 2.47$ | $27.53 \pm 1.45$ | $3.96 \pm 1.26$ | $5.59 \pm 0.87$ |
| TRADES | PACS | $91.16 \pm 0.08$ | $79.89 \pm 0.22$ | $79.70 \pm 0.95$ | $72.32 \pm 0.77$ | $57.96 \pm 1.56$ | $57.63 \pm 1.45$ |
| | OfficeHome | $69.12 \pm 0.15$ | $54.52 \pm 0.74$ | $56.14 \pm 0.59$ | $48.47 \pm 0.45$ | $35.79 \pm 1.14$ | $36.11 \pm 1.63$ |
| | VLCS | $78.58 \pm 0.17$ | $63.01 \pm 0.79$ | $63.30 \pm 0.63$ | $69.36 \pm 0.68$ | $52.78 \pm 1.66$ | $53.27 \pm 0.97$ |
| | TerraIncognita | $69.81 \pm 0.43$ | $58.64 \pm 0.79$ | $59.38 \pm 0.99$ | $25.27 \pm 3.16$ | $5.49 \pm 0.59$ | $7.84 \pm 0.65$ |

training in the source domain results in provably better representations for fine-tuning on the target domain. Such seemingly contradictory findings can be reconciled by considering the key differences between transfer learning and domain generalization summarized in Table 1. Specifically, previous works in transfer learning assume that the model can sample the target domain at some point to perform fine-tuning. Since domain generalization does not allow access to the target domain, such benefits are not guaranteed. *We encourage future works to investigate under what conditions adversarial training helps the generalization accuracy with no fine-tuning on the target.*

**Q2: Does a higher source domain robustness correspond to a higher target domain robustness?** As expected, DNNs lose some robustness when evaluated on a target domain that is distinct from the training domains. This is evident by comparing the Robust Acc. (AA) and Robust Acc. (PGD) of the source domain to those of the target domain in Table 2. For example, the PACS model trained with PGD augmentation experiences a drop of 19.31% in terms of AutoAttack robust accuracy and 22.30% in terms of PGD robust accuracy. However, it is consistently observed that ❷ **a higher robustness in the source domain corresponds to higher robustness in the target domain.** Our results suggest that one way to increase the out-of-distribution robustness of a deployed model is to improve its robustness in the source validation set, which supports the applicability of ongoing efforts in adversarial robustness research (Zhang et al., 2019; Wang et al., 2020; Wu et al., 2020). This is also validated by comparing the TRADES results with those of PGD in Table 2. TRADES, a stronger adversarial training method compared to PGD augmentation, achieves higher source domain robust accuracies and, in turn, higher target domain robust accuracies.

**Q3: Does the robustness-accuracy trade-off generalize to unseen domains?** As observed in Table 2, ❸ **a more robust model comes at the cost of standard accuracy not only in the source domain, but also in the target domain.** In most cases, both PGD and AutoAttack robust accuracies of standard-trained (baseline) models are around 0%. Those models, on the other hand, are the models that achieve the best in terms of generalizability, *i.e.* in terms of clean accuracy. However, the most robust models, i.e. TRADES, tend to have the lowest target domain clean data accuracy. Therefore, *consistent with the robustness literature (Tsipras et al., 2019), robustness comes at the cost of standard accuracy even in the unseen target domains.*

# 5 Certified Robustness and Domain Generalization

In Section 4, we analyzed the interplay between empirical robustness (obtained by adversarial training) and domain generalization. While empirical robustness studies give hints about the reliability of a given model under adversarial attacks, they give no guarantees against the existence of such adversaries. To deploy DNNs in dynamic environments (Koh et al., 2021), we need robustness guarantees to carry over into unseen domains. To that end, we study the generalizability of the certified robustness of DNNs. We also study whether the visual similarity between the source and target domains influences this robustness generalizability. We deploy Randomized Smoothing (RS) and DeformRS to certify DNNs against input perturbations and deformations, and we utilize the FID and R-FID metrics to compute cross-distribution similarity. We start by giving a brief overview of RS and DeformRS, followed by some background on FID and R-FID.

## 5.1 Background and Setup

**Certifying Against Additive Perturbations and Input Deformations.** Randomized smoothing (RS) (Cohen et al., 2019) is a method for constructing a "smooth" classifier from a given classifier $f_\theta$. The smooth classifier returns the average prediction of $f_\theta$ when the input $x$ is subjected to additive Gaussian noise:

$$g_\theta(x) = \mathbb{E}_{\epsilon \sim \mathcal{N}(0,\sigma^2 I)} \left[ f_\theta(x + \epsilon) \right]. \tag{7}$$

Let $g_\theta$ predict the label $c_A$ for input $x$ with some confidence, $i.e.$ $\mathbb{E}_\epsilon[f_\theta^{c_A}(x + \epsilon)] = p_A \geq p_B = \max_{c \neq c_A} \mathbb{E}_\epsilon[f_\theta^c(x + \epsilon)]$, then, as shown by Zhai et al. (2020), $g_\theta$'s prediction is certifiably robust at $x$ with certification radius:

$$R = \frac{\sigma}{2} \left( \Phi^{-1}(p_A) - \Phi^{-1}(p_B) \right), \tag{8}$$

where $\Phi^{-1}$ is the inverse CDF of the standard Gaussian distribution. As a result of Eq. 8, $\arg\max_i g_\theta^i(x+\delta) = \arg\max_i g_\theta^i(x), \forall \|\delta\|_2 \leq R$.

While Eq. 8 provides theoretical guarantees for robustness against additive perturbations, DNNs are also brittle against simple input transformations such as rotation. Alfarra et al. (2022a) extended randomized smoothing to certify parametric input deformations through DeformRS, which defined the parametric smooth classifier for a given input $x$ with pixel coordinates $p$ as follows:

$$g_\phi(x,p) = \mathbb{E}_{\epsilon \sim \mathcal{D}} \left[ f_\theta(I_T(x, p + \nu_{\phi+\epsilon})) \right], \tag{9}$$

where $I_T$ is an interpolation function ($e.g.$ bilinear interpolation) and $\nu_\phi$ is a parametric deformation function with parameters $\phi$ ($e.g.$ $\nu$ is a rotation function and $\phi$ is the rotation angle). Analogous to the RS formulation in Eq. 7, $g$ outputs the average prediction of $f_\theta$ over deformed versions of the input $x$. Alfarra et al. (2022a) showed that parametric-domain smooth classifiers are certifiably robust against perturbations to the parameters of the deformation function. In particular, $g$'s prediction is constant with certification radius:

$$\begin{aligned} R &= \sigma \left( p_A - p_B \right) && \text{for } \mathcal{D} = \mathcal{U}[-\sigma, \sigma], \\ R &= \frac{\sigma}{2} \left( \Phi^{-1}(p_A) - \Phi^{-1}(p_B) \right) && \text{for } \mathcal{D} = \mathcal{N}(0, \sigma^2 I). \end{aligned} \tag{10}$$

Simply put, as long as the perturbations to the deformation function parameters ($e.g.$ rotation angle) are within $R$, the prediction of $g$ remains constant. In this work, we leverage RS and DeformRS to study the generalizability of certified robustness to unseen target domains.

**Measuring Perceptual Similarity between Source and Target Distributions.** The Fréchet Inception Distance (FID) is a widely used metric for evaluating generative models (Heusel et al., 2017). It provides a measure of similarity between the generated images and a reference dataset based on extracted feature representations. These representations are extracted using a pre-trained neural network, typically the Inception network (Szegedy et al., 2016). FID assumes that the Inception features of an image distribution $\mathcal{D}$ follow a Gaussian distribution with mean $\mu_\mathcal{D}$ and covariance $\Sigma_\mathcal{D}$, and it measures the $\ell_2$ Wasserstein distance between the two Gaussian distributions. Hence, $\text{FID}(\mathcal{D}_1, \mathcal{D}_2)$ can be calculated as:

$$\text{FID}(\mathcal{D}_1, \mathcal{D}_2) = \|\mu_1 - \mu_2\|^2 + \text{Tr}\left(\Sigma_1 + \Sigma_2 - 2(\Sigma_1 \Sigma_2)^{1/2}\right), \tag{11}$$

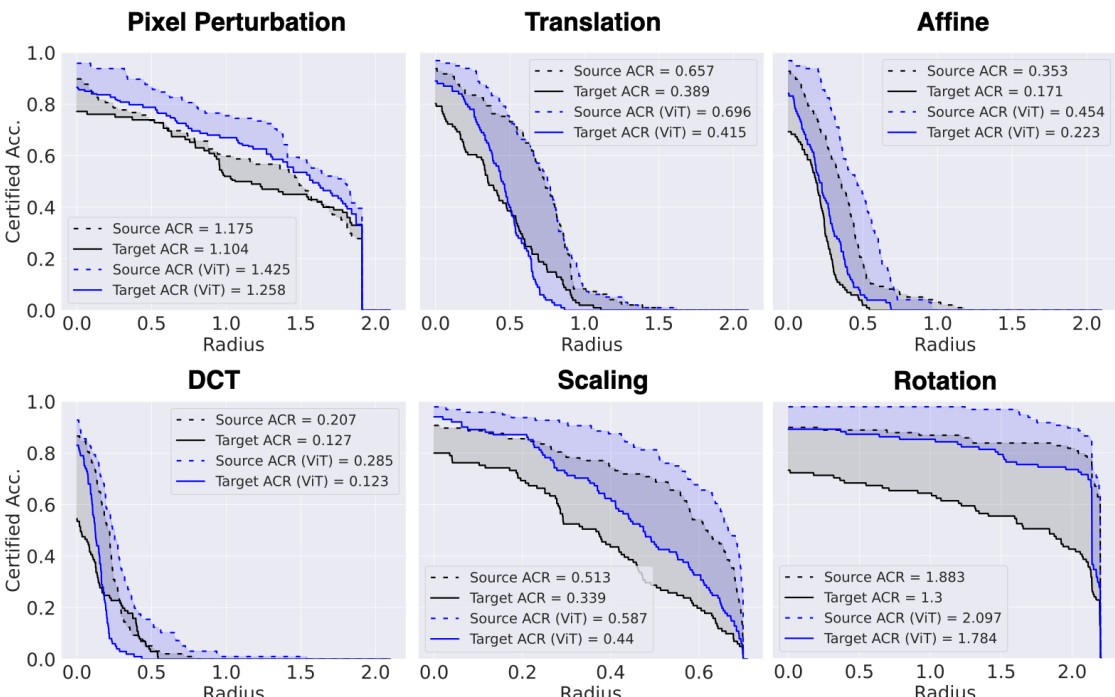

Figure 2: **Generalizability of certified robustness.** We certify ResNet-50 and ViT-Base against pixel perturbations and input deformations in the source and target domains of PACS. We observe that 1) certified robustness generalizes to unseen domains, and that 2) a stronger architecture (ViT-Base) leads to a better source and target certified accuracy.

where $\mathrm{Tr}(\cdot)$ is the trace operator. Note that the statistics of both distributions are empirically estimated from their corresponding image samples.

Recently, Alfarra et al. (2022b) demonstrated that quality measures for generative models, such as FID, can be vulnerable to adversarial attacks, leading to unreliable assessments of perceptual similarity between two distributions (Kynkäänniemi et al., 2023). In light of these findings, the authors proposed a robust variant of FID, called R-FID, that addresses this issue by replacing the standard Inception feature extractor with an adversarially trained Inception network, resulting in a more reliable metric for evaluating image quality.

Our experiments aim at capturing the interplay between visual perceptual similarity between source and target domains and the generalization of robustness. The notion of *perceptual similarity*, *i.e.* the degree to which two image distributions appear similar to humans, has been explored in evaluating generative models (Zhang et al., 2018; Shmelkov et al., 2018). Since the FID has been demonstrated to correlate with human judgment of image quality (Heusel et al., 2017; Lucic et al., 2018), we use it to compare different image domains and assess their perceptual similarity. Unlike the generative models literature that measures FID/R-FID between real and generated distributions of images, we measure the FID/R-FID between the source and target distributions which are two real image distributions. Our goal is to assess whether there is a correlation between the distributions' visual similarity and the models' ability to generalize to the unseen target distribution.

**Experimental Setup.** To split the data into source and target domains, we use the *Photo, Art, Cartoon,* and *Sketch* distributions from PACS (Li et al., 2017). We use RS to certify pixel perturbations and DeformRS to certify five input deformations: rotation, translation, scaling, affine, and a deformation characterized by a Discrete Cosine Transform (DCT) basis. Following (Gulrajani & Lopez-Paz, 2021), we employ data augmentation during training and train solely on the source domains. To evaluate the certified robustness of the trained classifier, we plot the certified accuracy curves for both the source and target domains for each

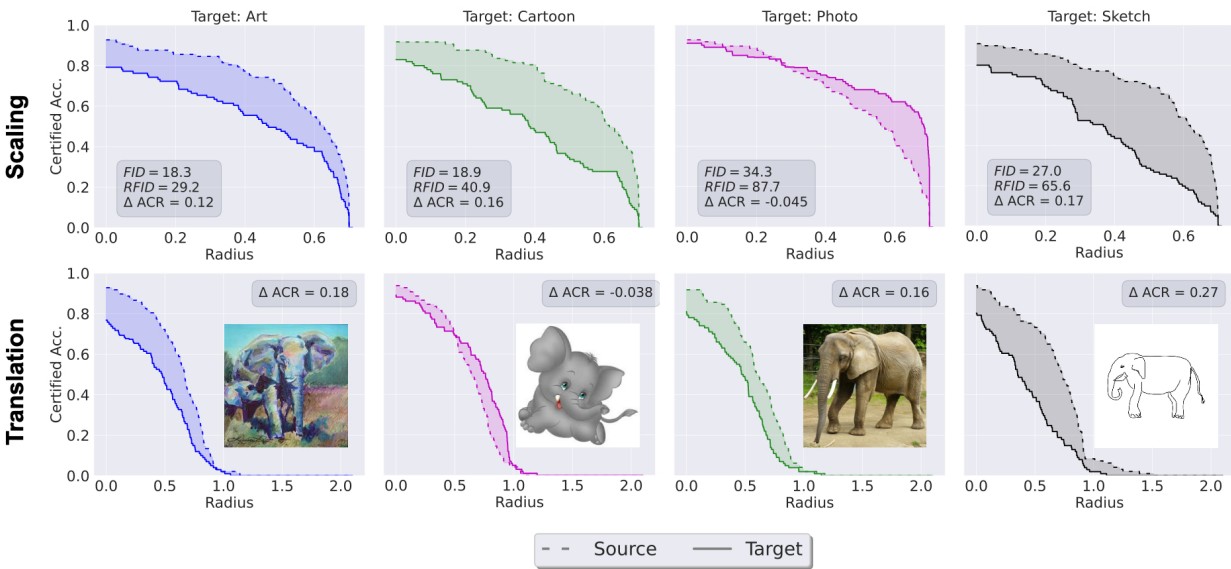

Figure 3: **Does visual similarity correlate with robustness generalizability?** We vary the target domain and plot the certified accuracy curves for two deformations: scaling and translation. A sample from each domain is shown in the second row. The FID/R-FID distances between the source domains and each target are reported in the first row. Visual similarity, measured by FID and R-FID, does not correlate with the level of robustness generalization to the target domain.

considered deformation. The certified accuracy at a radius $R$ is the percentage of the test set that is both classified correctly and has a certified radius of at least $R$. We calculate the certified radius for a given input through either Eq. 8 for pixel perturbations or Eq. 10 for input deformations. Here, we report the envelope plots, which illustrate the best certified accuracy per radius over all values of the smoothing deformation parameter $\sigma$. We leave the detailed results for each choice of $\sigma$ to the appendix. We employ Monte Carlo sampling with 100k samples and a probability of failure of $10^{-3}$ to estimate $p_A$ and bound $p_B = 1 - p_A$, following the standard practice (Zhai et al., 2020; Cohen et al., 2019; Alfarra et al., 2022a). Finally, we follow (Zhai et al., 2020) in reporting the Average Certified Radius (ACR) of correctly classified samples.

Regarding the architecture, we follow DomainBed (Gulrajani & Lopez-Paz, 2021) in selecting ResNet-50 as a backbone. To assess the effect of deploying a more powerful architecture on the generalizability to unseen domains, we also include experiments with the recent transformer model ViT-Base (Dosovitskiy et al., 2021).

## 5.2 Generalizability of Certified Robustness to Unseen Target Domains

We investigate when certified robustness generalizes to unseen domains. We first show how much Certified Accuracy (CA) is maintained when the target domain exhibits a distribution shift. Then, we study whether a stronger backbone architecture boosts the CA generalizability. Finally, we evaluate whether perceptual similarity, as measured by FID and R-FID (Heusel et al., 2017; Alfarra et al., 2022b), predicts the generalization of certified robustness. **Q4: Can certified robustness, obtained via randomized smoothing, generalize to unseen domains?** We train smooth classifiers on a collection of source domains and certify the models on both the source and target domains. The target domains are *unseen* before certification. We plot the source CA curve with dashed black lines and the target CA curve with solid textcolorblack in Figure 2, along with the corresponding ACR. Our results show that ❹ **a considerable portion of the certified robustness, acquired by randomized smoothing, is maintained in the unseen domain.** When certified against pixel perturbations in the unseen domain, the average certified radius of ResNet-50 drops by around 6% only. Utilizing DeformRS, we extend this result from pixel perturbations to geometric deformations, such as rotation. To explain these findings, we show in Table 9 in the appendix that invariance to different deformations, learned through data augmentation, generalizes to unseen distributions. Overall,

our results illustrate the importance of research efforts that improve on randomized smoothing (Zhai et al., 2020; Eiras et al., 2022). *To address real-world security challenges, we encourage future certified robustness works to conduct experiments on domain generalization datasets.*

**Q5: Does the target certified accuracy improve when the feature extractor is improved?** To investigate the influence of the backbone architecture on the certified robustness of a deployed model, we change the architecture from ResNet-50 to ViT-Base and plot the target CA curve in solid textcolorblack in Figure 2. We observe that the target ACR obtained by ViT-Base on PACS is higher than the target ACR obtained by ResNet-50 across deformations. ❺ **A significant improvement of the target certified robustness is achieved by using a stronger backbone.** This result is consistent with the robustness literature (Gowal et al., 2020), where stronger backbones exhibit better robustness, and the domain generalization literature (Gulrajani & Lopez-Paz, 2021), where stronger backbones exhibit better generalization accuracy. *We believe that research on better generalization can lead to better certified robustness in unseen domains.*

**Q6: Does the generalizability of certified robustness correlate with the perceptual similarity between the source and target domains?** In all previous experiments, we considered the average certified accuracy over all possible target domains. We now conduct a more fine-grained study to these target domains individually. We measure the drop in the average certified radius ($\Delta ACR$) between the source and target domains with the perceptual similarity between both domains captured by FID (Heusel et al., 2017) and the more robust R-FID (Alfarra et al., 2022b). To that end, we conduct experiments on PACS where we select one domain as the unseen target and treat the rest as source domains. We train a classifier on the source data and plot the certified accuracy curves against scaling and translation deformations on both the source and target domains in Figure 3 accompanied by $\Delta ACR$. We also report the FID and R-FID between the source and target domains. Note that *higher* FID/R-FID indicates *less* similarity of distributions. ❻ **Perceptual similarity, as captured by FID and R-FID, is not predictive of performance and robustness generalizability.** Surprisingly, the photo domain, which has the highest FID (34.3) and R-FID (87.7) scores, exhibits the largest certified accuracy generalization. In this case, the ACR for the target domain is higher than the source domain resulting in a negative $\Delta ACR$ ($-0.1$ when certifying against translation). The appendix includes experiments with other deformations where we observe similar behavior, along with potential reasons why the FID/R-FID are poor predictors of robustness generalization. *We regard the development of a suitable distribution similarity metric, which better correlates with the level of generalizability, as an important research direction.*

Table 3: **Evaluation on the Medical Dataset CAMELYON17** shows robustness generalization.

|  |  | Baseline | PGD | TRADES |
|---|---|---|---|---|
| **Source** | Clean Acc. | $97.48 \pm 0.10$ | $95.31 \pm 0.16$ | $93.59 \pm 0.18$ |
|  | Rob Acc. (AA) | $0.44 \pm 0.40$ | $83.33 \pm 1.80$ | $85.86 \pm 1.11$ |
|  | Rob Acc. (PGD) | $1.10 \pm 0.26$ | $84.23 \pm 1.38$ | $85.89 \pm 0.99$ |
| **Target** | Clean Acc. | $93.56 \pm 2.51$ | $91.41 \pm 1.78$ | $90.49 \pm 1.75$ |
|  | Rob Acc. (AA) | $0.10 \pm 0.17$ | $61.04 \pm 4.53$ | $67.58 \pm 3.97$ |
|  | Rob Acc. (PGD) | $0.88 \pm 0.75$ | $62.5 \pm 2.66$ | $66.31 \pm 7.69$ |

## 6 Real-world Application: Medical Images

To demonstrate the applicability of the DG setup to real-world settings, we investigate the generalization of robustness in medical diagnostics. Data collected by medical imaging techniques, like computed tomography (CT) and magnetic resonance imaging (MRI), is susceptible to noise. This noise includes intensity variations caused by subject movement (Shaw et al., 2019), respiratory motion (Axel et al., 1986), quantum noise associated with X-rays (Hsieh, 1998), and inhomogeneity in the MRI magnetic field (Leemput et al., 1999). Moreover, due to privacy concerns, a model trained in one medical institution should be deployed in another with limited data sharing and retraining (Kaissis et al., 2020; Ziller et al., 2021; Liu et al., 2021). To test

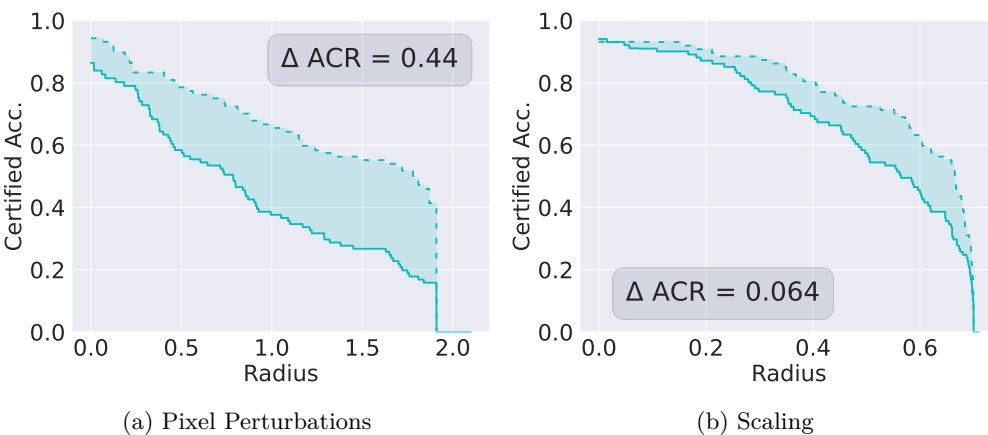

(a) Pixel Perturbations                     (b) Scaling

Figure 4: **Certified robustness in the medical domain.** The generalization of robustness against pixel variations in medical images is critical. Yet, there is a gap in certified robustness when the DNN is deployed in an unseen hospital.

the generalization of robustness in this practical setup, we use the DG dataset WILDS CAMELYON17 (Bándi et al., 2019; Koh et al., 2021) to train models on tissue images from four hospitals and evaluate them on images from an unseen hospital. For the first time, in addition to domain generalization, we explore robustness in CAMELYON17. The task in WILDS CAMELYON17 is to predict whether a tissue image contains a cancerous tumor or not.

**Adversarial Augmentation for Better Pixel Variation Generalizability.** In Table 3, we test the generalization accuracy of a standard-trained ($\epsilon = 0$) and robust ($\epsilon = 2/255$) models by following the setup from Section 4. Unlike the domains studied in Table 2, adversarial training maintains a high clean accuracy in the unseen target hospital. While the clean accuracy of the standard model is 93.5%, the clean accuracy of the PGD-trained model is 91.4%. This clean accuracy generalization may be attributed to the similarity between pixel perturbations and the underlying domain shift in the medical images. More importantly, the AutoAttack robust accuracy improves from 1% to 83% in the source hospitals, and from 0% to 61% in the unseen hospital. The results are similar for the TRADES-trained model, with even higher robustness generalization. *We encourage future works to study different adversarial training methods that go beyond pixel perturbations, and to propose application-specific augmentations for different distribution shifts.*

**Certified Robustness.** Next, we investigate the generalizability of certified robustness to the unseen hospital. We follow the setup in Section 5 and measure the certified accuracy on the source and target domains. We observe from Figure 4 that some of the certified robustness generalizes to the unseen hospital when evaluated with pixel perturbations and scaling deformations. We include the results for other deformations in the appendix. We note that the drop in certified accuracy to the unseen hospital (given pixel perturbations) is 4 times what we saw in the PACS dataset in Section 5. This is concerning, as many sources of noise affect medical imaging data, so robust medical diagnostics is important for real-world adoption of AI for Health. *We encourage future research to develop better methods to close the target-source gap in certified robustness.*

## 7    Conclusion

We conducted an extensive empirical analysis of the interplay between adversarial robustness and domain generalization. We found that empirical and certified robustness generalizes to unseen domains, including in a real-world setting. We also showed that visual similarity is not predictive of the level of generalizability. Based on our findings, we encourage more research on: (i) methods that improve certified accuracy in unseen domains, and (ii) distribution similarity metrics that correlate with the generalization accuracy.

## Acknowledgments

This work was supported by the King Abdullah University of Science and Technology (KAUST) Office of Sponsored Research (OSR) under Award No. OSR-CRG2021-4648 and the SDAIA-KAUST Center of Excellence in Data Science and Artificial Intelligence (SDAIA-KAUST AI).

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

## Appendix

## A    Empirical Robustness and Domain Generalization

### A.1    Testing Different Budget Norms

Table 4: **Evaluation of $\ell_\infty$ Robustness.**    We train models in the source domain and then evaluate their clean accuracy and robust accuracy in the source and target domains. The models are trained using either PGD adversarial training, or TRADES adversarial training. The robust accuracy is measured against AutoAttack and PGD adversarial attacks. For adversarial training and evaluation, the perturbation norm is set to $\epsilon = 4/255$. Overall, the robustness of PGD-trained and TRADES-trained models transfers to the target distribution.

| Method | Dataset | Source | | | Target | | |
|---|---|---|---|---|---|---|---|
| | | Clean Acc. | Robust Acc. (AA) | Robust Acc. (PGD) | Clean Acc. | Robust Acc. (AA) | Robust Acc. (PGD) |
| PGD | PACS | $90.68 \pm 0.27$ | $66.43 \pm 0.85$ | $67.95 \pm 0.76$ | $70.97 \pm 0.58$ | $45.48 \pm 1.85$ | $45.51 \pm 1.11$ |
| | OfficeHome | $68.23 \pm 0.12$ | $41.75 \pm 0.19$ | $43.13 \pm 0.65$ | $47.77 \pm 0.63$ | $25.22 \pm 1.15$ | $26.5 \pm 1.19$ |
| | VLCS | $77.07 \pm 0.18$ | $48.58 \pm 0.52$ | $50.14 \pm 1.15$ | $66.54 \pm 0.85$ | $36.69 \pm 2.20$ | $36.22 \pm 2.52$ |
| | TerraIncognita | $66.29 \pm 2.91$ | $45.89 \pm 6.38$ | $51.84 \pm 1.26$ | $26.15 \pm 3.26$ | $3.56 \pm 1.44$ | $5.83 \pm 2.77$ |
| TRADES | PACS | $66.71 \pm 0.36$ | $53.10 \pm 0.88$ | $53.32 \pm 1.01$ | $50.96 \pm 1.27$ | $35.23 \pm 1.28$ | $35.65 \pm 1.80$ |
| | OfficeHome | $56.94 \pm 8.76$ | $39.01 \pm 5.87$ | $40.93 \pm 6.97$ | $40.54 \pm 4.21$ | $26.44 \pm 0.92$ | $26.33 \pm 3.33$ |
| | VLCS | $75.69 \pm 0.30$ | $54.31 \pm 0.40$ | $55.52 \pm 0.89$ | $65.20 \pm 1.27$ | $42.48 \pm 0.98$ | $43.07 \pm 1.30$ |
| | TerraIncognita | $64.20 \pm 0.08$ | $52.07 \pm 0.81$ | $53.02 \pm 0.77$ | $27.97 \pm 1.22$ | $10.45 \pm 3.97$ | $12.04 \pm 2.77$ |

Table 5: **Evaluation of $\ell_\infty$ Robustness.**    We train models in the source domain and then evaluate their clean accuracy and robust accuracy in the source and target domains. The models are trained using either PGD adversarial training, or TRADES adversarial training. The robust accuracy is measured against AutoAttack and PGD adversarial attacks. For adversarial training and evaluation, the perturbation norm is set to $\epsilon = 8/255$. Overall, the robustness of PGD-trained and TRADES-trained models transfers to the target distribution.

| Method | Dataset | Source | | | Target | | |
|---|---|---|---|---|---|---|---|
| | | Clean Acc. | Acc. (AA) | Acc. (PGD) | Clean Acc. | Acc. (AA) | Acc. (PGD) |
| PGD | PACS | 86.36 | 48.24 | 52.83 | 64.06 | 29.00 | 32.92 |
| | TerraIncognita | 78.36 | 6.35 | 34.67 | 34.64 | 0.39 | 8.98 |
| TRADES | PACS | 85.11 | 58.43 | 59.85 | 60.45 | 32.81 | 36.60 |
| | TerraIncognita | 60.78 | 44.30 | 44.01 | 28.29 | 15.43 | 16.31 |

**Q1: Do adversarially robust models generalize better than their standard-trained counterparts?**    Again, the answer is no. Adversarially trained models tend to experience a drop in generalizability when compared to their standard-trained counterparts.

**Q2: Does a higher source-domain robustness correspond to a higher target-domain robustness?** As expected, the answer is still yes. As observed in Table 4, when we have a higher robustness in the source domain, we consistently observe a higher robustness in the target domain even for higher budget norm .

**Q3: Does the robustness-accuracy trade-off generalize to unseen domains?**    Yes, similar to what was observed in $\epsilon = 2/255$ experiments, the robustness-accuracy trade-off exists in unseen domains even for higher budget norms. Robustness in the target domain comes at the cost of clean accuracy.

## A.2 Empirical Robustness Ablations

In this section, we conduct various ablations to assess the impact of different hyperparameters in adversarial training on our observations.

First, we investigate the effect of the number of adversarial training steps. Tables 6 and 7 illustrate the results of decreasing the number of PGD and TRADES steps to 1, as well as increasing them to 10 for PACS and TerraIncognita datasets respectively. As expected, increasing the number of steps leads to higher robust accuracy, at the expense of clean accuracy.

Table 6: **Effect of Number of Adversarial Training Steps (PACS).** Increasing the number of adversarial training steps improves the model's ability to handle adversarial examples in both source and target domains, *i.e.* increases model robustness. However, this improvement comes at the cost of reduced accuracy on clean examples in both domains.

| Method | Steps | Source | | | Target | | |
|---|---|---|---|---|---|---|---|
| | | Clean Acc. | Acc. (AA) | Acc. (PGD) | Clean Acc. | Acc. (AA) | Acc. (PGD) |
| PGD | 1 | 94.55 | 59.90 | 84.20 | 78.51 | 42.29 | 63.12 |
| | 10 | 92.30 | 78.48 | 87.06 | 73.52 | 55.47 | 66.13 |
| TRADES | 1 | 93.95 | 66.99 | 84.72 | 77.38 | 47.17 | 65.02 |
| | 10 | 91.29 | 79.92 | 86.93 | 73.24 | 60.45 | 68.38 |

Table 7: **Effect of Number of Adversarial Training Steps (TerraIncognita).** Increasing the number of adversarial training steps improves the model's ability to handle adversarial examples in both source and target domains, *i.e.* increases model robustness. However, this improvement comes at the cost of reduced accuracy on clean examples in both domains.

| Method | Steps | Source | | | Target | | |
|---|---|---|---|---|---|---|---|
| | | Clean Acc. | Acc. (AA) | Acc. (PGD) | Clean Acc. | Acc. (AA) | Acc. (PGD) |
| PGD | 1 | 84.40 | 8.53 | 59.11 | 36.30 | 0.49 | 6.25 |
| | 10 | 69.66 | 58.33 | 65.59 | 22.78 | 5.96 | 14.75 |
| TRADES | 1 | 84.34 | 17.15 | 66.6 | 33.96 | 1.17 | 8.98 |
| | 10 | 68.81 | 58.63 | 64.42 | 26.08 | 12.79 | 18.07 |

Second, we examine the effect of changing the value of $\beta$ on TRADES. By definition, increasing the value of $\beta$ places a greater emphasis on robustness. Therefore, it is expected that increasing $\beta$ to 6.0 would result in an increase in robust accuracy, but a decrease in clean accuracy. In contrast, a decrease of $\beta$ to 1.0 is expected to have the opposite effect, with a decrease in robust accuracy and an increase in clean accuracy. The results are summarized in Table 8.

Table 8: **Effect of Changing $\beta$ on TRADES.** By definition, increasing the value of $\beta$ means putting a higher weight on robustness. Therefore, as expected when we $\beta$ increase to 6.0 the robust accuracy increases, whereas the clean accuracy drops and vice versa for $\beta = 1.0$.

| Dataset | $\lambda$ | Source | | | Target | | |
|---|---|---|---|---|---|---|---|
| | | Clean Acc. | Acc. (AA) | Acc. (PGD) | Clean Acc. | Acc. (AA) | Acc. (PGD) |
| PACS | 1.0 | 92.33 | 78.91 | 77.90 | 75.73 | 57.23 | 59.12 |
| | 6.0 | 90.02 | 80.50 | 79.65 | 71.09 | 58.11 | 60.75 |
| TerraIncognita | 1.0 | 78.21 | 51.82 | 53.58 | 27.70 | 3.42 | 3.12 |
| | 6.0 | 65.55 | 59.02 | 57.29 | 26.11 | 13.38 | 15.14 |

# B  Certified Robustness and Domain Generalization

### B.1  How is it possible for randomized smoothing to to certify models in unseen domains?

To understand why randomized smoothing is able to certify models in unseen domains, we investigate the hypothesis that the learned invariance to noise and geometric deformations generalizes to unseen domains. To that end, we train models in the source domain and then evaluate their accuracy on deformed samples from the source and target domains. During evaluation, the images are transformed with some $\epsilon$-parameterized deformation. Models that never see the corresponding deformation before evaluation time are denoted by "Without" in Table 9. Models trained on deformed images are denoted by "With". All the experiments are repeated four times with different seeds, and the average performance is reported. We observe that augmenting the source dataset equips the models with invariance against various deformations, and that this deformation invariance generalizes to the unseen target distribution. This invariance generalization is consistent with Figure 2, which illustrates the domain-generalizability of the certified robustness obtained through randomized smoothing.

Table 9: **Training with augmentations leads to domain-generalizable invariance to various deformations.** During evaluation, the images are transformed with some $\epsilon$-parameterized deformation. Models denoted by "With" are trained on augmented images transformed by the same deformation, and models denoted by "Without" do not see deformed images at training time. We note that the learned invariance against various deformations generalizes to the unseen target distribution.

| | | Without | | With | |
| --- | --- | --- | --- | --- | --- |
| **Deformation** | $\epsilon$ | **Source** | **Target** | **Source** | **Target** |
| **Gaussian Noise** | 0.1 | 87.98 | 75.13 | 93.36 | 79.60 |
| | 0.25 | 60.30 | 52.74 | 91.37 | 75.05 |
| | 0.37 | 38.99 | 36.90 | 89.06 | 72.49 |
| | 0.5 | 23.59 | 21.17 | 86.38 | 68.70 |
| **Affine** | 0.1 | 92.32 | 79.48 | 93.86 | 79.59 |
| | 0.3 | 75.07 | 61.70 | 87.85 | 68.44 |
| | 0.5 | 50.53 | 40.91 | 74.65 | 53.51 |
| | 0.7 | 35.99 | 28.37 | 63.51 | 42.14 |
| **DCT** | 0.1 | 81.06 | 66.53 | 91.48 | 72.52 |
| | 0.3 | 36.89 | 28.78 | 79.97 | 53.47 |
| | 0.5 | 23.69 | 19.42 | 70.02 | 41.51 |
| | 0.7 | 19.40 | 16.94 | 61.74 | 37.10 |
| **Rotation** | 0.1 | 92.71 | 77.96 | 94.03 | 79.02 |
| | 0.3 | 84.06 | 70.58 | 92.14 | 74.92 |
| | 0.5 | 75.25 | 61.27 | 90.36 | 70.19 |
| | 0.7 | 68.98 | 57.76 | 89.32 | 62.10 |
| **Scaling** | 0.1 | 93.96 | 79.68 | 94.25 | 80.63 |
| | 0.3 | 92.77 | 78.80 | 94.01 | 77.97 |
| | 0.5 | 90.26 | 76.52 | 92.91 | 78.34 |
| | 0.7 | 85.74 | 72.27 | 90.75 | 73.95 |
| **Translation** | 0.1 | 93.91 | 81.06 | 94.53 | 81.15 |
| | 0.3 | 91.98 | 78.32 | 93.37 | 78.76 |
| | 0.5 | 86.72 | 73.18 | 90.09 | 73.70 |
| | 0.7 | 77.87 | 65.39 | 84.94 | 66.65 |

### B.2 The effect of $\phi$ on the generalization of certified robustness

To complement Figure 2, we investigate the behavior when the deformation parameter $\phi$ varies. Following Section 5.2, we certify ResNet-50 and ViT-Base against pixel perturbations and input deformations in the source and target domains of PACS. We break down each envelope curve in Figure 2 into multiple curves, each representing one choice of $\phi$ in Eq. 9. We label each curve with the corresponding $\phi$ value in Figure 5. We observe that the effect of $\phi$ largely depends on the type of perturbation. On the one hand, for Scaling and Pixel Perturbations, a higher $\phi$ values corresponds to a larger Average Certified Radius (ACR); on the other hand, for Affine, DCT, and Translation, a higher $\phi$ values might correspond to a smaller ACR. This is because for the latter group of deformations, a higher $\phi$ results in a completely deformed image, which hinders the certification ability of the model, even at a small radius. We visualize images from these deformations in Section D. Note that the trends in Figure 2 still stand. Specifically, (1) for $\phi$ values where there's a reasonable certified accuracy in the source, that certified accuracy generalizes to the target. Moreover, (2) A stronger architecture (ViT-Base) generally leads to a better source and target certified accuracy.

### B.3 Does visual similarity correlate with robustness generalizability?

We repeat the experiments in Section 5.2, which aim to evaluate the ability of FID/R-FID to predict the generalization of robustness, on pixel perturbations and the following deformations: rotation, affine, and DCT. We observe from Figure 6 that the FID/R-FID values do not predict the level of generalizability of certified robustness, which matches our paper findings.

**Why does visual similarity not correlate with the level of robustness generalizability?** The FID and R-FID metrics rely on two fundamental assumptions. First, they assume that the deep features extracted from ImageNet pre-trained networks follow a multivariate Gaussian distribution (Heusel et al., 2017). Second, they assume that the extracted features correlate with human perceptual judgment (Zhang et al., 2018). Visual similarity, measured by these metrics, and robustness generalizability are not always correlated in our experiments. While surprising, this discrepancy has several possible reasons. Firstly, the approximation of Inception features as multivariate Gaussians is an oversimplified assumption that may not capture the complexities of each image distribution, as noted in previous work (Shmelkov et al., 2018). Secondly, The use of an ImageNet pretrained Inception module to extract features for each image distribution may not be representative of the considered image distributions in the domain generalization literature. Moreover, the use of ImageNet pre-trained Inception might introduce its own biases, as observed in (Kynkäänniemi et al., 2023). Lastly, deep neural networks tend to be biased towards texture, while humans are biased towards shape (Geirhos et al., 2019), which suggests that perceived visual similarity may not always align with DNN performance on classification tasks, as we also find in our experiments.

While our work sheds some light on the interplay between visual similarity and robustness generalization, more research should be done to disentangle the correlation between different components of the distribution similarity metric and the generalization. For instance, it is unclear how different choices for the feature extractor or distance metric used to compute the FID or R-FID may impact the correlation with generalization performance. Moreover, other metrics that better capture the image distribution's subtleties, possibly based on both shape and texture, may provide a more consistent correlation with generalization.

## C Real-world Application: Medical Images

We repeat the certified robustness experiments in Section 6 on the following deformations: affine, DCT, translation, and rotation. We observe from Figure 8 that the source-target certification gap is similar for affine, DCT, and translation. However, the certified accuracy curves for rotation are different. This makes sense when we consider the way the Camelyon17 dataset is constructed (Bándi et al., 2019; Koh et al., 2021). The dataset includes cropped histopathological images, each of which may contain a tumor tissue in the central 32x32 region. Due to the nature of this construction, rotated versions of the image look similar, which explains why the source and target certified radii remain almost constant. Samples from the Camelyon17 dataset are visualized in Figure 7.

## D Visualizing the domain generalization datasets

In Figure 12, we visualize a few samples from each of the domain generalization datasets we used in the paper. We note that the datasets are diverse in terms of the nature of domain shifts and real-world applicability. Along with the clean samples, we visualize deformed versions of the samples under various values of $\sigma$ for all the studied deformations.

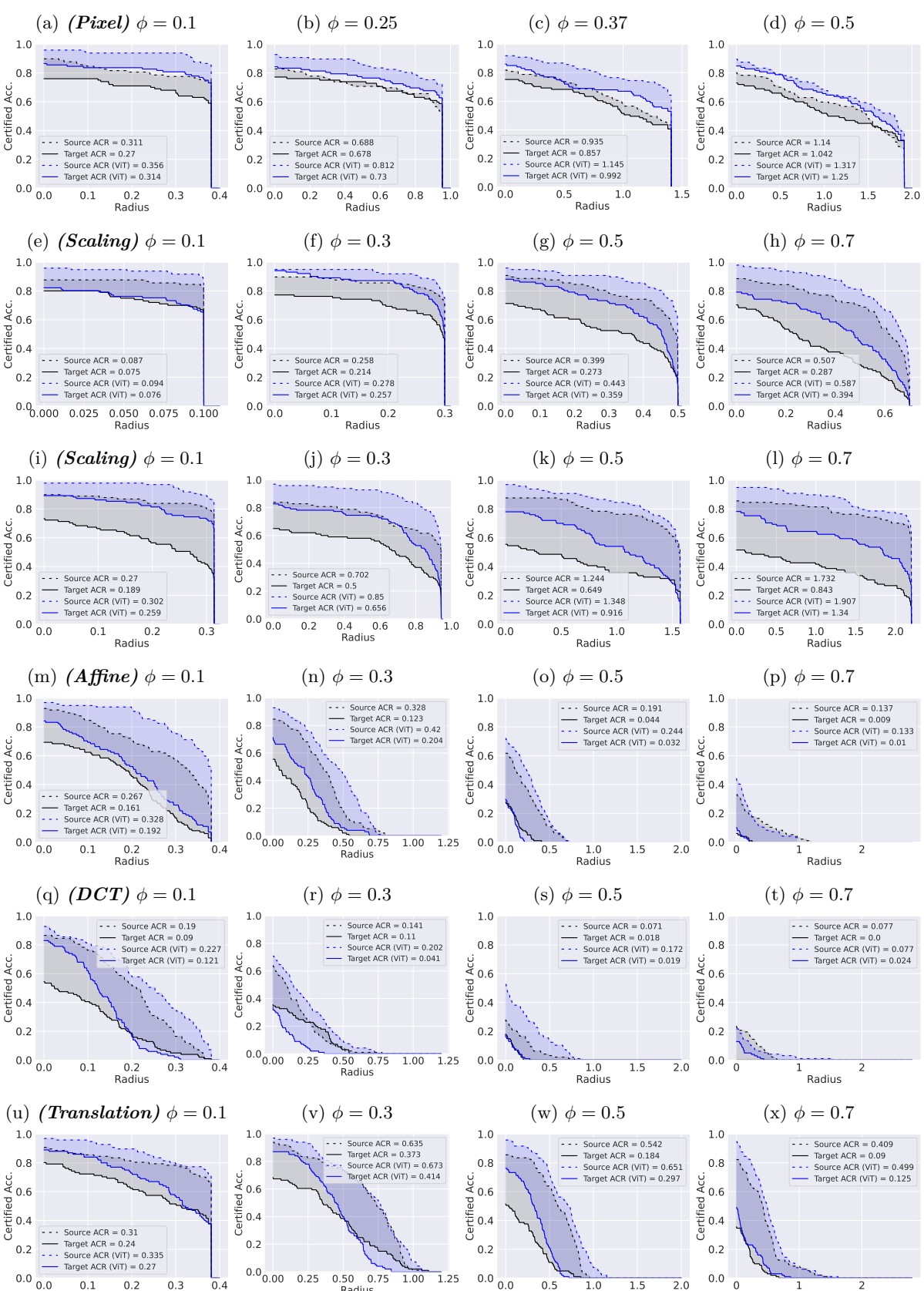

Figure 5: **Effect of Varying the Deformation Parameter** $\phi$**.** We observe that 1) for Pixel Perturbations, Scaling, and Rotation, the higher $\phi$ gets, the larger the Average Certified Radius (ACR) becomes; and (2) for Affine, DCT, and Translation, high $\phi$ values can result in low ACRs.

Figure 6: **Does visual similarity correlates with robustness generalizability?** We vary the target distribution and plot the certified accuracy curves for different deformations. The FID/R-FID distances between the source and target distributions are shown in the first row. Visual similarity (FID and R-FID) does not correlate with the level of robustness generalization to the target domain.

**Clean Samples**

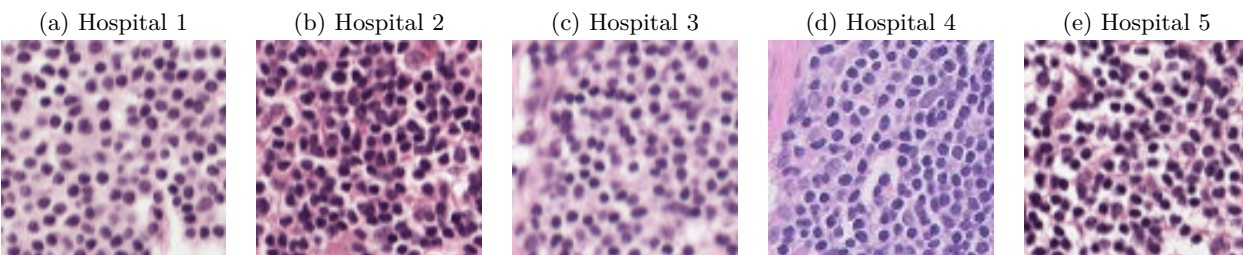

Figure 7: **A visualization of the images taken from the 5 hospitals in Camelyon17.**

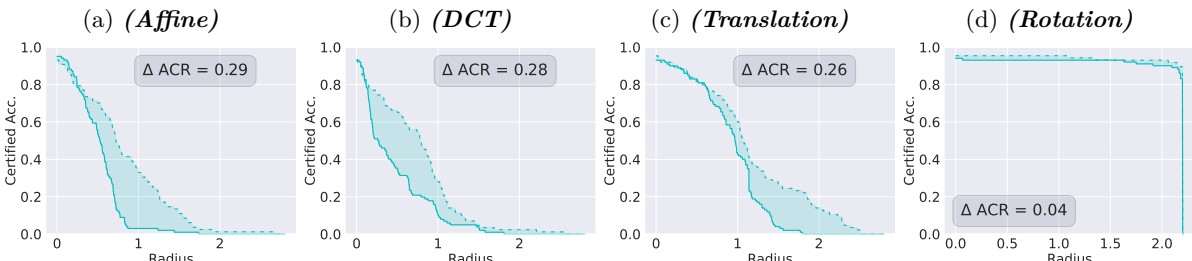

Figure 8: **Does visual similarity correlates with robustness generalizability?** We vary the target distribution and plot the certified accuracy curves for two deformations: scaling and translation. A sample from each distribution is shown in the second row. The FID/R-FID distances between the source distributions and each target are inset in the first row. Visual similarity, measured by FID and R-FID, does not correlate with the level of robustness generalization to the target domain.

**Clean Samples**

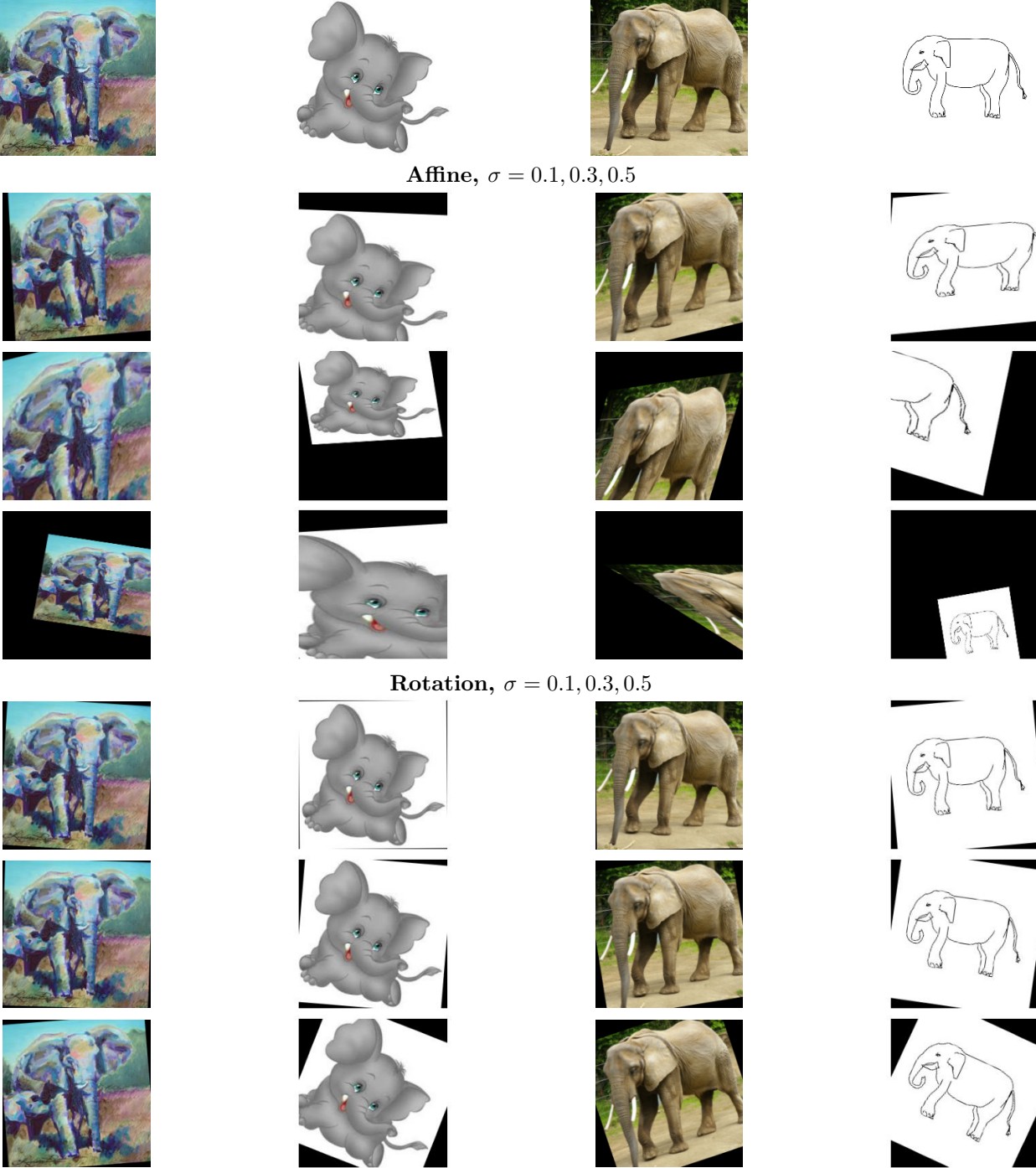

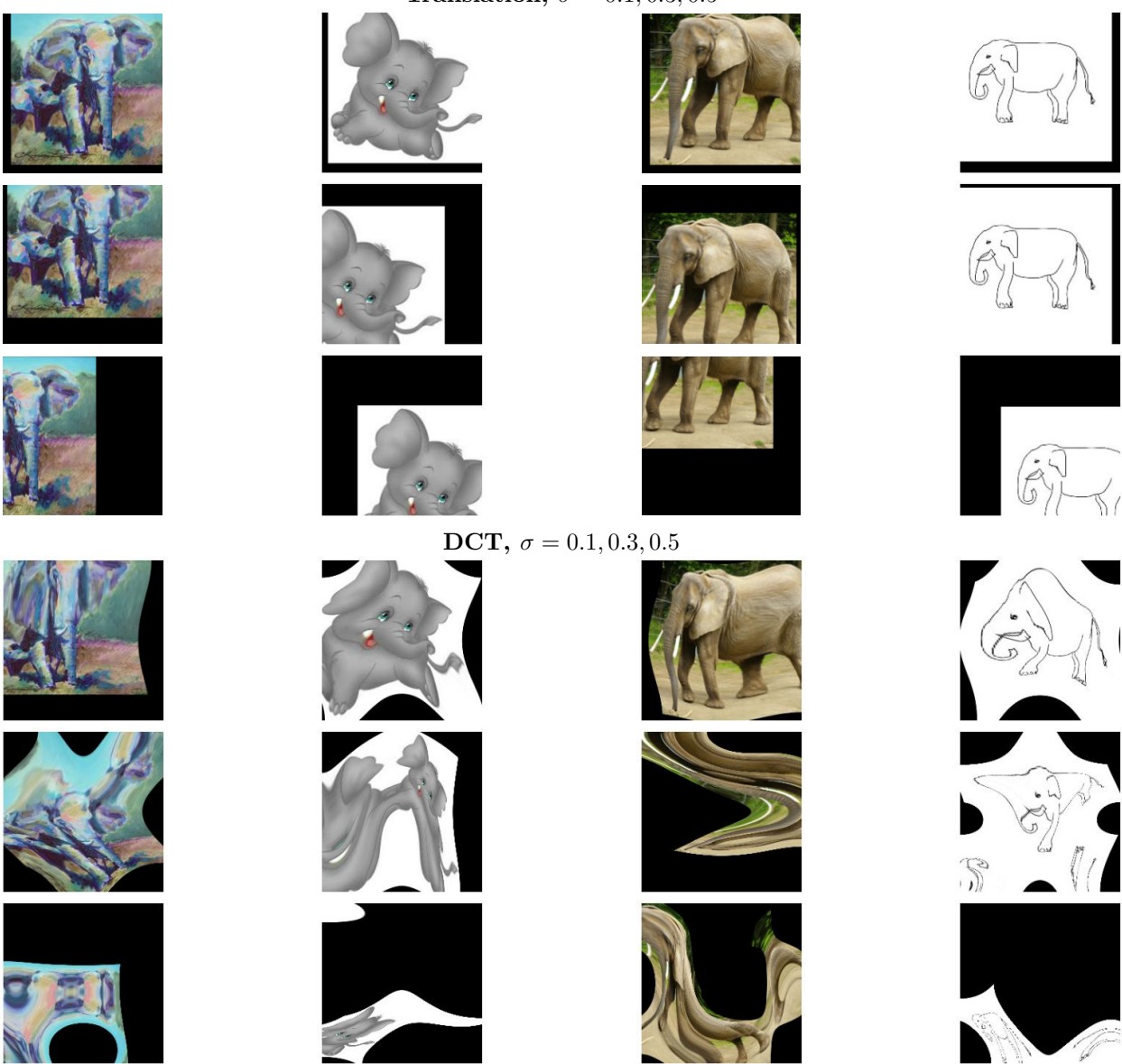

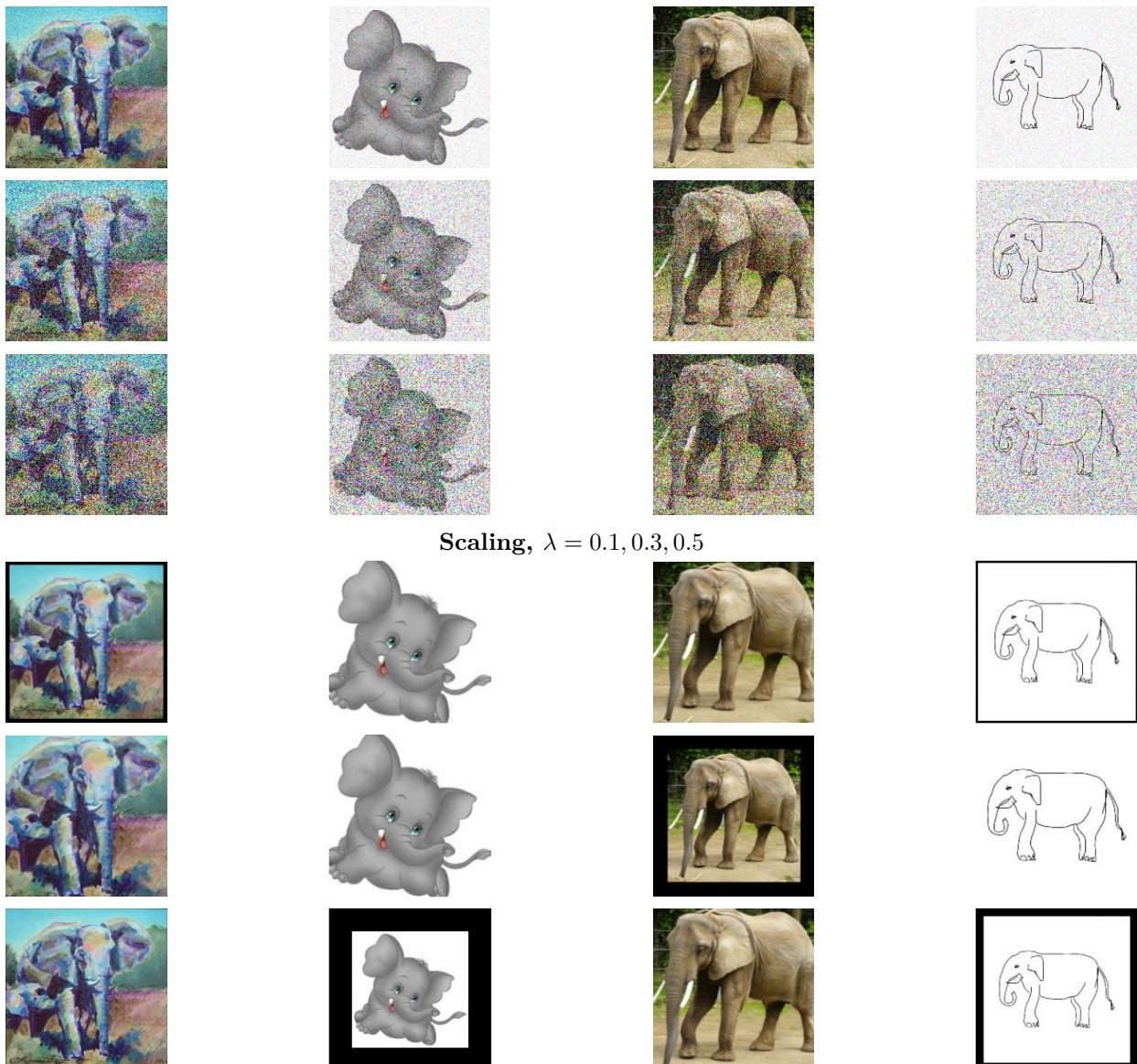

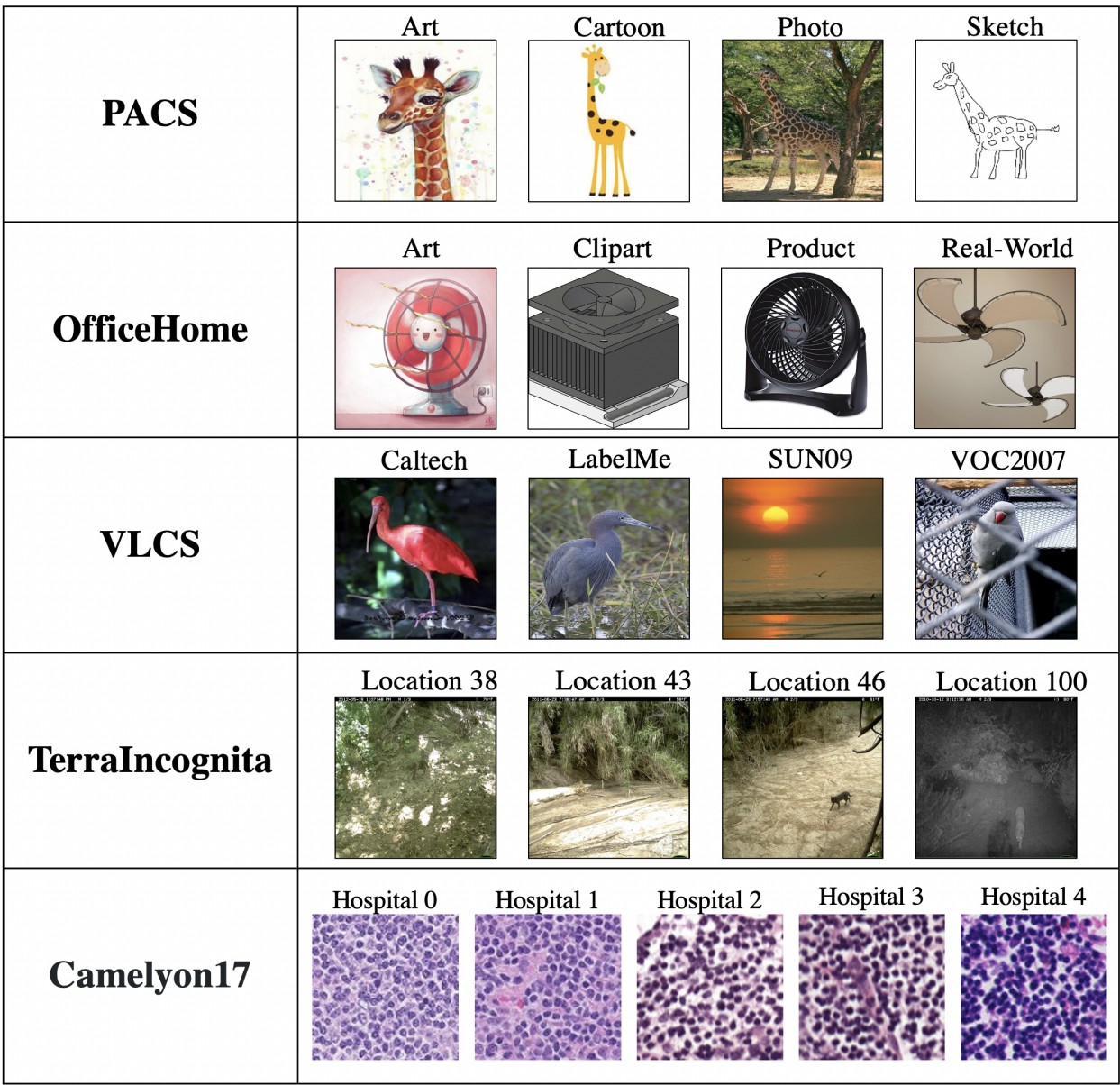

Figure 12: **Domains of Studied Datasets.** In our work we considered five datasets each consisting of multiple domains.

