# OpenReview forum: "Generalizability of Adversarial Robustness Under Distribution Shifts"
_TMLR — Accepted by TMLR_

### Review · Reviewer_RYco · 2023-03-27

**Summary Of Contributions:**

This paper empirically evaluates the generalization and adversarial robustness on target domains achieved by the models that are adversarially trained on source domains where the target domain is distinct from the source domain. By deploying both empirical and certified defence, the authors show that the adversarially-trained models can lead to higher robustness on unseen domains compared with standardly-trained models. Besides, without finetuning, the adversarially-trained models cannot result in higher standard test accuracy on unseen domains. Finally, the authors provide two interesting research directions based on their empirical findings.

**Audience:**

Yes

**Broader Impact Concerns:**

I have no broader impact concerns.

**Claims And Evidence:**

Yes

**Requested Changes:**

1. Could the authors provide more implementational details in Table 2? Actually, I am confused about the results in the column of “Source”. What is the source domain here? And what is the meaning of the results of “Source”?

2. Should the term $P_{XY}^{(T)}$ in Eq. (1) be $P_{XY}^{(|T|)}$?

3. I found that you use PGD-5 for adversarial raining and $\beta=3$ for TRADES. Normally, we use PGD-10 for adversarial raining and $\beta=6$ for TRADES, which follows the setting of the original paper.  Could you discuss the reason for your choice of these hyper-parameters, since this paper is an empirical study and the hyper-parameters could have a large impact on the final results? Possibly, you can provide the ablation studies on these critical hyper-parameters.


**Strengths And Weaknesses:**

Strengths:

+ This paper is well-written and well-organized. The motivation is clear. Previous studies focus on the interplay between adversarial robustness and transfer learning. This paper seems to for the first time study the interplay between adversarial robustness and domain generalization.
+ The authors conducted comprehensive experiments, which clearly demonstrate the interplay between adversarial robustness and domain generalization. The authors clearly state the key findings from their experiments, which could be interesting and meaningful for the community.

Weaknesses:

- Although comprehensive empirical studies have been done, the paper lacks some theoretical analyses for interpreting their empirical findings.

---

> ### Author Response · Authors · 2023-04-11
> **Response to requested changes.**
>
>
> We thank the reviewer for the insightful feedback and the proposed changes. We have addressed each requested change, which we hope will improve our paper. All the changes are highlighted in blue in the revised manuscript.
>
> **Regarding more implementational details in Table 2, including what the "Source" column represents.** For each considered dataset, we select a subset of $N-1$ domains to be the source (training) domains and keep the $N^{th}$ domain as the target (evaluation) domain. For example, we have four splits in PACS: Photo, Art, Cartoon, and Sketch. So we have four combinations of source vs. target domain splits. This process is repeated $N$ times. We follow DomainBed in reporting the average result across the $N$ different selections of the target domain. The whole process is run 3 times with different seeds, and the standard deviation is reported.
>
> With that in mind, the results in the “Source” column of Table 2 represent the performance of the trained model on the training distribution. To clarify, we divided the source domains (training domains) into two sets: a training subset (80%) and a validation subset (20%). It's important to note that the results we present in Table 2 are based on the validation subset.
> Reporting the “Source” results has two purposes: (1) Validating that the considered adversarial training schemes result in robust models when evaluated on the source domains. (2) Comparing the performance and robustness between the observed source domain, and the unseen target domain. Reporting these results allows us to assess the generalizability of empirical robustness of adversarially trained models to unseen domains.
>
>
> **Regarding Eq. (1).** We thank the reviewer for catching this typo. The term $P_{XY}^{(T)}$ should be replaced with ${P}_{XY}^{(N+1)}$, where $N+1$ is the index of the target domain. We have updated the text to mention this explicitly. We’ve also fixed the typo in other equations, including Eq. (5) and Eq. (6).
>
> **Regarding the choice of hyperparameters and providing more ablation results.** We thank the reviewer for the suggestion. The original choice for $\beta$ was made to maintain a good balance between clean accuracy and robust accuracy. As we increase $\beta$, we prioritize robustness over clean data performance. The choice of PGD-5 was done to reduce the computational cost as we ran a lot of experiments to verify our findings (Table 2 alone requires 144 experiments). Nonetheless, we run experiments with different adversarial training steps (1 and 10 steps) and different selection of $\beta$ for TRADES (1 and 6) and verify that our conclusions on the transferability of robustness still holds. The experiments are found in Appendix A.2. We observe that increasing $\beta$ for TRADES, while improving the generalizability against adversarial attacks, deteriorates the performance on clean data samples.

---

### Review · Reviewer_fkDt · 2023-03-31

**Summary Of Contributions:**

The paper investigates the generalizability of empirical and certified adversarial robustness to unseen domains in the context of domain generalization. The authors observe that both types of robustness generalize to unseen domains, and that the level of generalizability does not correlate well with input visual similarity. They also extend their study to cover a real-world medical application, in which adversarial augmentation significantly boosts the generalization of robustness with minimal effect on clean data accuracy.

**Audience:**

Yes

**Claims And Evidence:**

Yes

**Requested Changes:**

1) It may be necessary to provide clearer descriptions of the evaluation criteria for empirical robustness generalization. Some of the conclusions mentioned above require more reasonable explanations.
2) It would be helpful to have a clearer description of some of the experimental setups, such as the number of attack steps and perturbation radius used by AA and PGD in the empirical robustness experiments.
3) The paper presents validation experiments for perturbation radii of epsilon=2/255 and 4/255, but I am curious about how the robustness generalization would perform under larger perturbation radii.


**Strengths And Weaknesses:**

Strengths:
1) This paper systematically investigates the robustness generalization under distribution shift scenarios. The authors discuss both empirical and certified robustness, which is comprehensive.
2) The writing of the paper is friendly to readers. The authors' description of the background and problem setting is clear, including the statement of the difference between domain generalization and transfer learning.
3) The experimental design and results of the paper validate most of the arguments. The experiments in real-world application is interesting.

Weaknesses:
1) The main argument of the paper, "robustness can generalize to unseen domains," is not surprising to me because adding adversarial perturbations is already a form of a specific distribution shift. It is easy to imagine that a model that is robust to adversarial perturbations can partially handle other types of distribution shifts (domain generalization).
2) I am confused by some of the conclusions. For example, in Q1 of section 4.2, how does the author define "generalize better"? And why does the author say that the TRADES model generalizes worse than the PGD model when it looks like they both have similar performance loss in the target domain? In Q3, how is "TRADES model tends to be the least generalizable" defined?

---

> ### Author Response · Authors · 2023-04-11
> **Response to requested changes.**
>
> **Regarding the definition of “generalize better”.** We thank the reviewer for raising this point. We understand the source of the confusion. The generalization in the referenced statements refers to "clean data generalization", which is determined by the target clean accuracy in Table 2. This is measured in the third rightmost column in Table 2. In Transfer learning, adversarial training in the source domain results in a higher clean data accuracy in the target domain. In Q1, we wanted to highlight that this is not the case for domain generalization.
> To avoid the confusion, we've rewritten the question as follows:
> _“Q1: Does adversarial training result in a better generalization to clean samples in the target domain?”_
>
> We've also changed the conclusion of that question to the following:
> _“Unlike transfer learning, where robust training in the source domain is favorable, robust training does not improve the clean data accuracy in the target domain if no fine-tuning is allowed.”_
>
> Regarding Q3: TRADES tends to be the least generalizable since it obtains the lowest clean accuracies on the target domain (lower than both PGD and Baseline). In Q3, we changed the referenced sentence to state:
> _“However, the most robust models, i.e. TRADES, tend to have the lowest target domain clean data accuracy.”_
>
> **Regarding a clearer description of the experimental setups.** as mentioned in the “Adversarial Training Setup” paragraph in Sec 4.1,  we set the perturbation radius to 2/255 (paper) and 4/255 (Appendix). We also specified parameters such as $beta$ and the number of adversarial training steps.
> Additionally, we report the number of adversarial steps for evaluation in a newly added paragraph in Section 4.1 called “Evaluation Setup”. For PGD, we use 20 steps for evaluation, whereas for AutoAttack, we use the default number of steps for each method. We hope these details and our code (which we plan to release) will allow for the full reproducibility of our results.
>
> **Regarding how the robustness generalization would perform under larger perturbation radii.** We thank the reviewer for this suggestion. We conducted experiments with $\epsilon=8/255$ and followed similar experimental setups to Section 4.2 in Appendix A.1.
> Overall, the robustness of PGD-trained and TRADES-trained models transfers to the target distribution. Note that the larger $\epsilon$ is, the more the clean accuracy on both the source and target domains drops.

---

### Review · Reviewer_1R7v · 2023-04-06

**Summary Of Contributions:**

In this paper, the authors studied the generalization ability of adversarial robustness from multiple source domains to one unseen target domain under the distribution shift. Specifically, the work shows the generalization ability of both adversarial training (called empirical robustness in the paper) and certified robustness from the source to the target domain without any target domain information. The work also includes lots of empirical evaluations and interesting empirical findings.

**Audience:**

Yes

**Broader Impact Concerns:**

None.

**Claims And Evidence:**

Yes

**Requested Changes:**

1. Briefly present more information about the FID and R-FID scores and the reasons why use them as the evaluation of the visual similarity.
2. Add more discussions about the point 2 mentioned in the Weaknesses.
3. Check if the use of the illustration images in Figure 3 is proper or not, and remove the watermark if it is permitted to be used in this paper.

**Strengths And Weaknesses:**

Strengths:
1. The research problem, adversarial robustness under distribution shift, is very practical and important for deploying many real-world applications.
2. The work contains lots of empirical studies and findings, which would be interesting for the community.
3. The paper is well-written and easy to understand, the illustrations of the problem setting is clear, and the empirical analysis is also well presented.

Weaknesses:
1. Why use FID and R-FID score as the evaluation of the visual similarity? It would be better to explain the reasons, and also briefly present what is FID and R-FID in the paper.
2. Why visual similarity captured by FID and R-FID does not correlate with the level of robustness generalization
to the target domain? Any possible interpretations about this? Would this point still be true if using other scores to measure the similarity? More discussions about this point would be interesting.
3. There is a clear watermark on the elephant image in Figure 3 (i.e., in the 2nd row and 2nd column). Is there also a copyright issue about using this image? Are the shown images the real images used in the target domain?

---

> ### Author Response · Authors · 2023-04-11
> **Response to requested changes.**
>
> **Regarding background information on the FID and R-FID.** The Fréchet Inception Distance (FID) is a widely used metric for evaluating generative models. It provides a measure of similarity between the generated images and a reference dataset based on extracted feature representations. These representations are extracted using a pre-trained neural network, typically the Inception network.
>
> FID assumes that the Inception features of an image distribution $\mathcal D$ follow a Gaussian distribution with mean $\mu_\mathcal D$ and covariance $\Sigma_\mathcal D$, and it measures the $\ell_2$ Wasserstein distance between the two Gaussian distributions.
> Hence, $\text{FID}(\mathcal D_1, \mathcal D_2)$ can be calculated as:
> \begin{equation}
>     \text{FID}(\mathcal D_1, \mathcal D_2) = \| \mu_1 - \mu_2\|^2 + \text{Tr}\left(\Sigma_1 + \Sigma_2 - 2(\Sigma_1  \Sigma_2)^{\frac{1}{2}}\right),
> \end{equation}
>
>
> Our experiments aim to capture the interplay between visual perceptual similarity and the generalization of robustness. The notion of ”perceptual similarity”, i.e. the degree to which two image distributions appear similar to humans, has been explored in evaluating generative models [A, B]. Since the FID metric has been demonstrated to correlate well with human judgments of image quality, we use it to compare different image domains and assess their perceptual similarity. Specifically, we measure the FID/R-FID between the source and target distributions. Our goal is to assess whether there is a correlation between the distributions' visual similarity and the models' ability to generalize to the unseen target distribution.
>
> Full background information about the FID (and its more robust version R-FID) is added to the “Measuring Perceptual Similarity between Source and Target Distribution” paragraph in Sec 5.1.
>
> **Regarding why visual similarity captured by FID and R-FID does not correlate with the level of robustness generalization to the target domain.** We agree that exploring why visual similarity does not correlate well with the level of robustness generalizability is interesting. We provide the following thoughts on this, but we believe this is an interesting direction that deserves its own paper.
>
> Note that the FID and R-FID metrics rely on two fundamental assumptions. First, they assume the deep features extracted from ImageNet pre-trained networks follow a multivariate Gaussian distribution. Second, they assume that the extracted features correlate with human perceptual judgment.
> Visual similarity, measured by these metrics, and robustness generalizability are not always correlated in our experiments.
>
> While surprising, this discrepancy has several possible reasons. Firstly, the approximation of Inception features as multivariate Gaussians is an oversimplified assumption that may not capture the complexities of each image distribution, as noted in previous work [B]. Secondly, Using an ImageNet pre-trained Inception module to extract features for each image distribution may not represent the considered image distributions in the domain generalization literature. Moreover, using ImageNet pre-trained Inception might introduce its own biases, as observed in [C].
> Lastly, deep neural networks tend to be biased towards texture, while humans are biased towards shape [D], which suggests that perceived visual similarity may not always align with DNN performance on classification tasks, as we also find in our experiments.
>
> While our work sheds some light on the interplay between visual similarity and robustness generalization, more research should be done to disentangle the correlation between different components of the distribution similarity metric and the generalization. For instance, it is unclear how different choices for the feature extractor or distance metric used to compute the FID or R-FID may impact the correlation with generalization performance. Moreover, other metrics that better capture the image distribution's subtleties, possibly based on both shape and texture, may provide a more consistent correlation with generalization.
>
> Following the reviewer's suggestion, this discussion has been added to Appendix C.3.
>
> **Regarding the watermark on the used image.** Thank you for spotting this. The watermark is actually from the original dataset samples. However, to avoid potential copyright issues, we replaced the figure with another sample from the dataset.
>
> [A] Zhang, Richard, et al. “The unreasonable effectiveness of deep features as a perceptual metric,” CVPR 2018
>
> [B] Shmelkov, Konstantin, Cordelia Schmid, and Karteek Alahari. "How good is my GAN?" ECCV 2018
>
> [C] Kynkäänniemi, Tuomas et al. “The Role of ImagNet Classes in Fréchet Inception Distance”, ICLR 2023
>
> [D] Geirhos, Robert et al. "ImageNet-trained CNNs are biased towards texture; increasing shape bias improves accuracy and robustness," ICLR 2019

---

### Author Response · Authors · 2023-04-11
**Common comments to all reviewers.**

We thank all the reviewers for their insightful feedback. We would like to highlight that we modified our manuscript in accordance with the received comments where all the modifications are in blue. We summarize our modifications as follows (each item includes the reviewer that raised the point):
* Updated the notation in Section 3 and fixed the typo in equations 1, 5 and 6. **[RYco]**
* Added a new paragraph to Sec 4.1  titled “Evaluation Setup”, which now includes more complete details about the evaluation procedure and parameters followed in Table 2. **[RYco]**
* Rephrased parts of Q1 and Q2 in Section 4.2 to better differentiate between generalization in terms of clean data performance and generalization of adversarial data performance.  **[fkDt]**
* Added more background information on FID and R-FID in Section 5.1. **[1R7v]**
* Added a discussion on why visual similarity may not correlate with robustness generalization in Section B.3 of the Appendix. **[1R7v]**
* Included additional experiments: (1) Ablation to study the effect of number of PGD and TRADES adversarial training steps (Appendix A.2); (2) Ablation to study the effect of the hyperparameter \beta of TRADES (Appendix A.2); (3) Testing our conclusions under higher perturbation norm (Appendix A.1). **[RYco, fkDt]**
* Modified Figure 3 to replace the image with a watermark with another image from DomainBed. **[1R7v]**

---

### Decision · Action_Editors · 2023-05-04

**Recommendation:** Accept as is

**Comment:**

The paper studied (empirical and certified) adversarial robustness under distribution shifts. This is an important yet underexplored research problem. The authors raised interesting questions and found inspiring answers with supportive evaluations, as the first step towards the goal of understanding how adversarial robustness transfers under distribution shifts. The paper received quite positive reviews from both adversarial-training and distribution-shift experts. Thus, we should definitely accept it for publication, and I think it deserves a Feature Certification.

**Audience:**

Yes.

**Claims And Evidence:**

Yes, very nicely.

---

> ### Author Response · Authors · 2023-05-19
> **Thank you for the valuable feedback and recognition**
>
> We would like to express our sincere gratitude to the reviewers and editor for their thoughtful reviews and invaluable feedback on our paper. Your comments and suggestions have significantly enhanced the quality of our work. We are thrilled that our paper has been well-received and honored with the 'Featured Certification.'
>
> We would also like to inform you that the camera-ready version of the paper has been uploaded. Thank you all for your valuable contributions to the publication process.